# Using Transfer Learning Method to Develop an Artificial Intelligence Assisted Triaging for Endotracheal Tube Position on Chest X-ray

**DOI:** 10.3390/diagnostics11101844

**Published:** 2021-10-06

**Authors:** Kuo-Ching Yuan, Lung-Wen Tsai, Kevin S. Lai, Sing-Teck Teng, Yu-Sheng Lo, Syu-Jyun Peng

**Affiliations:** 1Professional Master Program in Artificial Intelligence in Medicine, College of Medicine, Taipei Medical University, Taipei 10675, Taiwan; traumayuan@gmail.com; 2Department of Surgery, DA CHIEN General Hospital, Miaoli 36052, Taiwan; 3Department of Medicine Education, Taipei Medical University Hospital, Taipei 110301, Taiwan; lungwen@tmu.edu.tw; 4Division of Critical Care Medicine, Department of Emergency and Critical Care Medicine, Taipei Medical University Hospital, Taipei 110301, Taiwan; kevinsl79@hotmail.com (K.S.L.); b101090152@tmu.edu.tw (S.-T.T.); 5Institute of Biomedical Informatics, Taipei Medical University, Taipei 110301, Taiwan

**Keywords:** artificial intelligence, endotracheal tube, chest X-ray, transfer learning

## Abstract

Endotracheal tubes (ETTs) provide a vital connection between the ventilator and patient; however, improper placement can hinder ventilation efficiency or injure the patient. Chest X-ray (CXR) is the most common approach to confirming ETT placement; however, technicians require considerable expertise in the interpretation of CXRs, and formal reports are often delayed. In this study, we developed an artificial intelligence-based triage system to enable the automated assessment of ETT placement in CXRs. Three intensivists performed a review of 4293 CXRs obtained from 2568 ICU patients. The CXRs were labeled “CORRECT” or “INCORRECT” in accordance with ETT placement. A region of interest (ROI) was also cropped out, including the bilateral head of the clavicle, the carina, and the tip of the ETT. Transfer learning was used to train four pre-trained models (VGG16, INCEPTION_V3, RESNET, and DENSENET169) and two models developed in the current study (VGG16_Tensor Projection Layer and CNN_Tensor Projection Layer) with the aim of differentiating the placement of ETTs. Only VGG16 based on ROI images presented acceptable performance (AUROC = 92%, F1 score = 0.87). The results obtained in this study demonstrate the feasibility of using the transfer learning method in the development of AI models by which to assess the placement of ETTs in CXRs.

## 1. Introduction

Mechanical ventilation is a life support modality commonly used in intensive care units (ICUs) for a wide range of situations, from scheduled surgical procedures to acute organ failure [1]. Suitable management of mechanical ventilation based on individual pathophysiology and responses to therapy can greatly improve outcomes [2]. Mechanical ventilation requires an artificial connection between the ventilator and the patient’s airway, involving tracheostomy, a jet needle, or most commonly an endotracheal tube (ETT) [3]. The suitability of tube placement must be confirmed as soon as possible after ETT intubation in order to minimize the risk of adverse events, such as tube dislodgement, aspiration pneumonia, tracheal injury, sinusitis, injury to vocal cords, stenosis of the trachea, or cuff over inflation. A number of methods have been developed to confirm ETT placement using a stethoscope, end-tidal CO2 levels, or portable chest X-rays (CXRs). Portable CXRs are currently the gold standard to confirm ETT placement, due to the fact that they are highly informative, inexpensive, and immediately available at the patient’s bedside in any location of the hospital [4]. Unfortunately, many clinicians lack expertise in CXR interpretation and are therefore unable to evaluate tube placement or identify situations indicative of potential ETT complications. Furthermore, the overwhelming workload of experienced radiologists often delays the preparation of formal reports. This situation has led to the development of various point-of-care methods to facilitate the timely assessment of tube placement [5].

The term artificial intelligence (AI) refers to data processing software that interacts with the world via feedback. AI is widely used to facilitate the interpretation of medical images [6,7], process biomedical signals (e.g., heart dysrhythmia from wearable sensors) [8,9], and facilitate disease prediction [10,11]. AI makes it possible to apply logical reasoning to data at scales that are too vast for the human mind to comprehend. The scaling-up of logical reasoning enables clinicians to leverage an enormous volume of medical knowledge in real-time and extends our knowledge by bringing together research in diverse disciplines [12]. Deep learning is a form of AI that has proven particularly effective in the processing of medical images of the chest, breast, brain, musculoskeletal system, and abdomen and pelvis [13]. AI can also be used to support clinical decisions, thanks to the availability of data from electronic health records and advancements in computation. For example, AI has been applied to CXRs to enable the early diagnosis of pneumonia, facilitate the selection of proper antibiotics, and estimate the likelihood of recovery. The ability of machine learning and neural networks to handle enormous volumes of data has also led to changes in clinical decision-making processes [14], such as the automated interpretation of medical images.

Deep learning is increasingly being used to automate the detection of thoracic disease in CXRs [15]. One such approach is the convolutional neural network (CNN), in which computational models are used to extract features from image data at various levels of abstraction using multiple processing layers. Note that CNNs establish models primarily from raw data, i.e., without the need for the manual extraction of features. In one previous study, deep learning proved at least as effective as senior radiologists in the detection of lung nodules in CXRs and computed tomography scans [16]. Deep learning has also proven effective when applied to other clinical conditions, such as the differentiation of interstitial lung disease, the segmentation of lung lesions, and the prediction of patient outcomes [16].

In the current study, we established a transfer learning-based AI system by which to assess the placement of ETTs for mechanical ventilation, with the aim of reducing the workload of medical staff and improving patient safety.

## 2. Materials and Methods

### 2.1. Data Sources

The development of the AI model in this study was based on the retrospective analysis of CXR images obtained from ICU patients at the Taipei Medical University Hospital (TMUH) during the period from January 2019 to June 2020. A total of 4293 CXRs were obtained from 2868 patients. Note that many ICU patients are subjected to CXR examinations several times during a single ICU stay. The CXRs labeled “CORRECT” or “INCORRECT” by three board-certificated intensivists in accordance with the placement of the ETT were considered ground truth observations. All CXRs in this study were saved in Joint Photographic Experts Group (JPEG) format (initial image size: 2800 × 1810–4238 × 3480 pixels). The study was approved (17 July 2020) by the TMUH-Joint Institutional Review Board of Taipei Medical University (TMU-JIRB No.: N202007011), and the need for written informed consent was waived due to the retrospective monocentric design of the study.

### 2.2. Image Preprocessing and Data Augmentation

After labeling, the CXRs underwent manual cropping by intensivists to create region of interest (ROI) images, including four landmarks: the right clavicle head, the left clavicle head, the carina, and the tip of the ETT (Figure 1). We also performed data augmentation during image cropping with a slight left or right deviation of the cropped image to increase the number of ROI images (1.5% more than the original FULL images). The two groups of images (FULL and ROI) were further split into training (TRAIN), validation (VAL), and test (TEST) datasets, respectively (Table 1). The training dataset was used to develop the algorithm, the validation dataset was used for model selection, and the test dataset was used to assess model accuracy when encountering images for which it had not been previously trained.

### 2.3. Transfer Learning

Image classification was achieved via transfer learning using the Tensorflow framework (Tensorflow 1.4, Google LLC, Mountain View, CA, USA) and the Keras library (Keras v 2.12, https://keras.io, accessed on 25 July 2020) to train the networks. We employed pre-trained deep learning models, including VGG16, Inception V3, ResNet50, and DenseNet169. In addition, VGG16 with Tensor Projection Layer (TPL) as well as a proprietary three-block CNN with TPL was also applied. Training a CNN from the beginning can be difficult when there is a lack of computing power and/or when using a dataset of limited size (no training dataset can be large enough). Transfer learning is meant to overcome these shortcomings. Two types of transfer learning are commonly used to deal with problems related to imaging processing. The first uses a pre-trained model for feature extraction to train a linear classifier for the new task. The second retrains a fully-connected layer atop a CNN, while fine-tuning the weights of the pre-trained network via back propagation. Due to the limited computing power and small dataset in the current study, we opted for the first approach.

### 2.4. Coding

The proposed Python code structure included three modules for sequential implementation: (1) algorithm-related, (2) image processing, and (3) model fitting and performance evaluation. A flowchart detailing the Python code structure and implementation can be found in Appendix A. When a pre-trained model is first loaded, the top argument can be set to “False” to prevent the loading of fully-connected output layers used to make predictions. This makes it possible to add and train a new output layer capable of providing binary classification results. The input tensor argument must be specified in accordance with the expected fixed size of the model input. A pre-trained model will not perform prediction without atop outputs activations directly from the previous convolution or pooling layer. A global pooling layer (e.g., max-global-pooling or average-global-pooling) can be used to summarize activations for use in a classifier task or as an input feature vector. The result is a vector used as an input feature descriptor [17]. The class argument specifies binary classes configured specifically for the output layer of our model. Note that we used the default settings for the pre-trained model and the Adam optimizer with the epoch number set at 20. Each algorithm was trained using FULL as well as ROI images.

Image file processing was first performed with the aim of ensuring that each image conformed to the requirements of the pre-trained model(s) to which it was applied. Keras’ ImageDataGenerator Class was used for pixel scaling prior to modeling. Under this scheme, the image dataset is wrapped, and images are returned to the algorithm in batches throughout the training, validation, and evaluation stages, after performing appropriate scaling operations. The ImageDataGenerator class provides a variety of pixel scaling methods as well as a range of data augmentation techniques. In the current study, we employed pixel standardization, in which scale pixel values have a zero mean and unit variance. Pixel standardization can be implemented at two levels: per-image (sample-wise) or per-dataset (feature-wise). Note that only the mean or the mean and standard deviation are required to standardize pixel values, regardless of whether the process is implemented sample-wise or feature-wise. The choice of pixel scaling scheme is determined by specifying arguments to the ImageDataGenerator class at the time when an instance is constructed [18].

The performance of the trained classifiers was evaluated using various combinations of features extracted using the pre-trained models. Note that each algorithm was trained using FULL and ROI images through 20 epochs. After model fitting, model performance was evaluated using the [model.evaluate] code in order to plot learning curves indicating loss and accuracy. After the model was established, predictions were performed using the TEST image set, the results of which were evaluated using the TEST annotation set as a reference. The metric “Test accuracy” refers to the performance of the model when using CXR images in the TEST Set (i.e., statistical defined accuracy). We plotted a confusion matrix comprising the prediction results and TEST annotation set to reveal the difference between the prediction result and the TEST set. We also evaluated the prediction performance of the trained models in terms of accuracy, recall, precision, F1 score, and area under receiver operating characteristic curve (AUC).

### 2.5. CXR Shape Profiles

CXR shape profiles were used to express the structural information extracted from each CXR image and to perform comparisons. CXR shape profiles have previously been used to obtain rough estimates of lung regions using histograms peak analysis and additional features regarding frontal/lateral CXR classification [19]. CXR shape profiles respectively illustrate (in the horizontal and vertical directions) the distribution of intensity values obtained by summing up pixel intensities in each column and row. Despite their simplicity, CXR shape profiles provide clear features representative of the image content, such that variations in the histogram are strongly correlated with the characteristics observed in the images.

### 2.6. Statistical Analysis

All statistical analysis was performed using the pROC package (version 1.7.3) in the R programming language (version 3.3.1; R Foundation, Vienna, Austria). The receiver operating characteristic (ROC), the area under the curves (AUC), and 95% confidence intervals were obtained where indicated. The DeLong non-parametric method was used to assess statistical differences among AUCs. A *p*-value of less than 0.05 was considered statistically significant.

## 3. Results

The FULL group included 4,293 image files and the ROI group included 4,354 files, after manual cropping. The FULL group included 2,580 “CORRECT” (CO) images (60.1%) and 1,713 “INCORRECT” (INCO) images (39.9%). In the ROI group, the CO ratio and INCO ratio was 61.7 and 38.3% (Figure 2). In terms of image size, the ROI group was significantly more heterogenous than was the FULL group. We created boxplots and distribution histograms to visualize patterns in the distribution of image file sizes in the three subsets (TRAIN, VAL, and TEST) of the FULL and ROI groups. Statistically significant differences were observed between the CO and INCO regarding image file size distribution in all three subsets of the ROI group but not of the FULL group (Figure 3). This may be attributed to the manual cropping used to create the ROI files. Images in the FULL group were retrieved from the PACS system (uniform image size). Despite establishing a protocol by which to implement manual cropping, there was considerable variation between cropped images in the ROI group.

Table 2 lists the performance of all 12 models. Note that 6 models were applied to the FULL image dataset and 6 were applied to the ROI image dataset. Overall, all of the algorithms performed better when applied to ROI images. We also plotted a performance dashboard for each model containing all statistic metrics, model structures, learning curves (accuracy and loss), confusion matrices, and ROC curves. The VGG16 algorithm trained using ROI images achieved the best performance (AUC = 92%), followed by VGG16 with TPL (AUC = 82%). A summary and diagram showing the architecture of the VGG16 used in this study (best performance) is included in the Appendix A. Figure 4 presents a complete performance report of VGG16 using ROI images. When applied to ROI images, VGG16 achieved validation accuracy (VAL_ACCURACY) of 0.82 and a validation loss (VAL_LOSS) of 0.49 during model fitting (Figure 4A). The learning curve is another model evaluation method used to detect problems, such as underfit or overfit, by plotting a given dataset (training, validation/evaluation) against a performance metric (loss, accuracy). The curve calculated from the training dataset (training learning curve) was used to evaluate how well the model learns (blue line in Figure 4B,C), whereas the curve plotted by a hold-out validation dataset (validation learning curve) was used to evaluate the model in terms of generalizability (orange line in Figure 4B,C). The Loss curves are calculated on the metrics by which the model is being optimized (Figure 4B), while the Accuracy curve is composed of the metric by which the model is evaluated and selected (Figure 4C). The learning curve of VGG16, when applied to ROI images, presented a good fit, as indicated by the gradual decrease in training loss and validation loss to the point of stability, with only a small gap between the two (Figure 4B). The accuracy curve presented a steady improvement throughout training (Figure 4C). After model fitting, we applied the trained VGG16 model to the TEST set plot and plotted a confusion matrix by which to estimate the precision, recall, and F1_score and accuracy (Figure 4A). Using the predictions generated by the trained VGG16 model in conjunction with the corresponding TEST dataset, we plotted the ROC and calculated the AUC. The trained VGG16 model achieved a high AUC of 0.92 (Figure 4D), indicating good model performance.

The three other pre-trained models failed to provide satisfactory performance, with AUROC values in the range of 46–57%. All four of the pre-trained models presented severe overfitting. We also implemented the pre-trained models using various numbers of epochs (3, 5, 10, and 20). The optimal numbers of epochs were as follows: DenseNet121 and ResNet50 (3 epochs), Inception V3 (5 epochs), and VGG16 (10 epochs).

## 4. Discussion

In this study, we developed an AI-based triage system to assess the placement of ETTs in CXRs. Four pre-trained frameworks (VGG16, Inception V3, ResNet50, and DenseNet169) were selected for model development. We also developed a novel CNN with a tensor projection layer (CNN_TPL) and VGG16 with a tensor projection layer (VGG16_TPL). Analysis was performed on original CXR images (FULL) and manually cropped CXR images (ROI). Model fitting was performed using 20 epochs.

The development of AI systems for object detection and recognition in X-rays is an emerging field, and many such studies have used transfer learning. Unlike disease diagnosis, the use of AI for object detection focuses mainly on the therapeutic tube and catheter [20,21,22]. In a similar study on the placement of feeding tubes, pre-trained deep CNN models (Inception V3, ResNet50, and DenseNet121) achieved AUC values of 0.82–0.87, which is far lower than that of the models developed for disease diagnosis [22]. Another study achieved 99.6% accuracy (95% confidence interval (CI): 97.5 to 100.0%) in identifying the manufacturer of a device from a radiograph and 96.4% (95% CI: 93.1 to 98.5%) accuracy in identifying the model group [20]. Overall, the ability of the network to identify the device manufacturer significantly exceeded that of any cardiologist (*p* < 0.0001 compared with the median human identification; *p* < 0.0001 compared with the best human identification).

The performance of the pre-trained models in detecting and evaluating the placement of ETTs did not meet our expectations. In particular, the carina and tip of the ETT can be difficult to identify in CXRs generated from the portable anteroposterior devices used in this study, which are prone to noise interference and tend to be lower in contrast resolution than standard posterior-anterior CXR devices. In the current study, image noise was classified into three distinct types. The first form of image noise was related to the constituent material of the ETT. In our facility, it is not uncommon to use non-kinking ETTs with a wire embedded within the tube to enhance rigidity, which tends to generate image noise. Note that we observed a considerable difference between plastic tubes and non-kinking tubes (Figure 5A). The second type of image noise was related to patient characteristics, including neck length, jaw location, clavicle orientation, and lung field condition. The patients in this study varied considerably in terms of age, size, weight, position, and disease; however, the image frame was not adjusted (one size fits all). These discrepancies greatly hindered the model’s development (Figure 5B). The third type of image noise was associated with the imaging environment, which often included other tube/catheters and heart-associated devices, many of which generated signals similar to those of the ETT. For example, many patients undergoing heart surgery have a wire over the sternum and/or prostheses, such as metallic valves, located very close to the ETT and causes noise (Figure 5C).

Note that the pre-trained models in this study were implemented using default settings with a new output layer to output binary results. It is very likely that performance could be improved through the tuning of hyperparameters, such as the learning rate (i.e., the degree to which the model changes in response to the estimated error associated with updates to model weights). Stochastic gradient descent is an optimization algorithm used to estimate the error gradient for the current state of a model using examples from the training dataset [23]. The weights of the model are then updated through the back propagation of errors, where the learning rate indicates the degree to which the weights are updated during training. The learning rate is a configurable hyperparameter presented as a small number between 0.0–1.023. In a previous study in which a deep CNN was used to detect body parts in CXR images, identification performance was improved by tuning the learning rate [21]. In that study, validation accuracy was only 50% when the learning rate was 0.01; however, it increased to 99% when the learning rate was adjusted to 0.001. Optimizing the learning rate is not a trivial matter, as an excessively small value could result in a lengthy training process, whereas an excessively large value could interfere with the learning of weights or destabilize the training process [23]. In the current study, we opted not to tune the learning rate because our initial objective was to employ only default settings. For all of the pre-trained models in this study, we employed the Adam optimization algorithm, which is an extension to stochastic gradient descent broadly adaptable to deep learning applications in computer vision and natural language processing [24]. The Adam optimizer combines the advantages of two other extensions of stochastic gradient descent: (1) the adaptive gradient algorithm (AdaGrad), which maintains a per-parameter learning rate to improve performance when using sparse gradients; and (2) root mean square propagation (RMSProp), which maintains per-parameter learning rates based on the average recent magnitudes of the gradients for a given weight [24]. The Adam optimizer is easily configurable, and the default configuration is well suited to most problems without any tuning. Nonetheless, TensorFlow documentation recommends a certain amount of tuning [24].

In this study, the VGG16 with a TPL achieved the second-best result (AORUC = 82%). In a regular CNN model, pooling layers are used to summarize features extracted from convolutional layers, which means that its main role is to reduce feature dimensionality. Note that the TPL is a more sophisticated dimension reduction layer (i.e., using tensor decomposition) [25], which considers subsequent multilinear mapping to a lower-dimensional feature space with output tensors. Note also that this is similar to multilinear principal component analysis (MPCA). Moreover, the TPL preserves the tensor structure while providing supervised dimension reduction. TPL uses label information (supervised) whereas MPCA does not (unsupervised) [25]. We believe that the good performance of VGG16 with TPL can be attributed to the effects of dimensionality reduction provided by TPL, which helped to alleviate interference associated with image noise.

Transfer learning using pre-trained models based on non-medical images (ImageNet) tends to outperform untrained models [26,27,28]. The first layer in most pre-trained networks is used to detect general features (e.g., edges and lines), as seen in architectures employing multiple neural networks, regardless of the type of training data [29]. The last layer in most pre-trained networks is specific to the training data. It should be possible to improve accuracy by leveraging the use of pre-trained weights for the initial layers in a well-formed neural network while reserving training mainly for the last layer (set to the initialization of random weights). Rajkomar et al. demonstrated that pre-training using grayscale images from ImageNet could improve accuracy beyond that achieved via pre-training with color images, as long as transfer learning is used for training [26]. Note however that this assertion is valid only if all of the layers except for the last layer are frozen. Another study demonstrated that high accuracy could be achieved using models pre-trained on color images at a reduced learning rate, even if the layers are not frozen (fine-tuning of all layers) [22]. In the current study, none of the layers were frozen; however, the algorithm was set to learn more slowly.

This study faced a number of limitations that should be considered in the interpretation of results. First, the dataset was inherently imbalanced, with a CO:INCO ratio of roughly 6:4. The lack of improperly placed ETTs in this study may be explained by the fact that a number of practical protocols had already been established for intubation. Clinicians are required to check the placement of the ETT using a stethoscope immediately after intubation and adjust the tube immediately if mispositioning is suspected. The second limitation is related to the transfer learning method. We used pre-trained models based on ImageNet, which should be satisfactory as long as the target images match those in the database. Unfortunately, most of the 14 million images in ImageNet are from daily life, with very few medical images and almost no CXRs. The noise and high degree of similarity among CXRs make it very difficult to find a well-designed pre-trained model appropriate to this application. Most of the pre-trained models in the current study achieved better performance when applied to ROI images; however, even the current ROI image is still too noisy. A new method for image cropping including only the carina and ETT tip, could further reduce image noise. We found that the image cropped with this new method contained only 0.87% of the FULL image, but still provided sufficient information to differentiate correct and incorrect positioning of the ETT. Further research will be conducted to assess the feasibility of using the narrow-cropping method to create source images.

## 5. Conclusions

This study demonstrates the feasibility of using pre-trained models in developing a transfer learning-based AI system by which to automate the assessment of ETT placement in CXRs. Most of the pre-trained models performed better when using ROI images. The best performance was achieved using VGG16 in conjunction with ROI images (AUC = 92%, F1 score = 0.87). Excessive image noise and inadequate hyperparameter tuning were the major causes of unsatisfactory performance. In such cases, TPL dimensionality reduction was shown to improve performance. Overall, these preliminary results demonstrate the feasibility of using transfer learning-based AI for the assessment of ETT placement in CXRs.

## Figures and Tables

**Figure 1 diagnostics-11-01844-f001:**
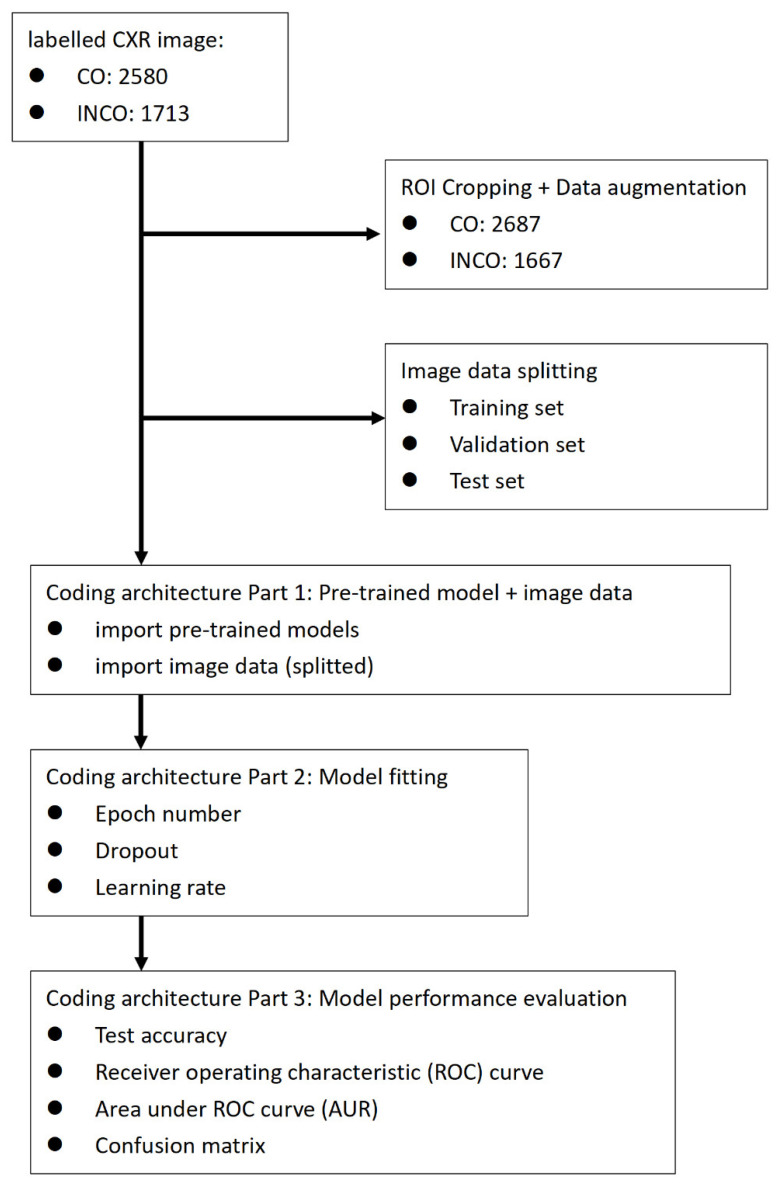
Flowchart of image data preparation, coding architecture, and evaluation of model fitness. CO: correct position of endotracheal tube. INCO: incorrect position of endotracheal tube.

**Figure 2 diagnostics-11-01844-f002:**
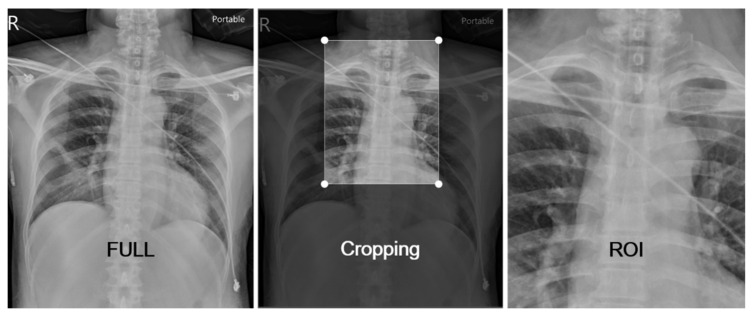
Manual cropping to produce ROI images, including the four landmarks (right clavicle head, left clavicle head, carina, ETT tip).

**Figure 3 diagnostics-11-01844-f003:**
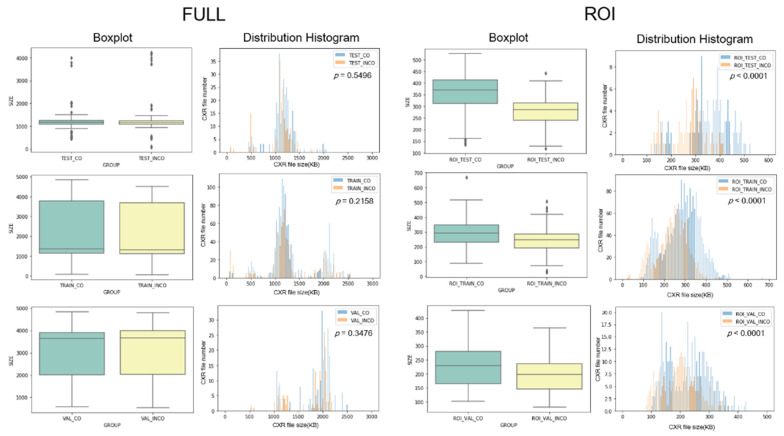
Boxplot and distribution histograms of CXR file size in the three subgroups of the FULL and ROI groups.

**Figure 4 diagnostics-11-01844-f004:**
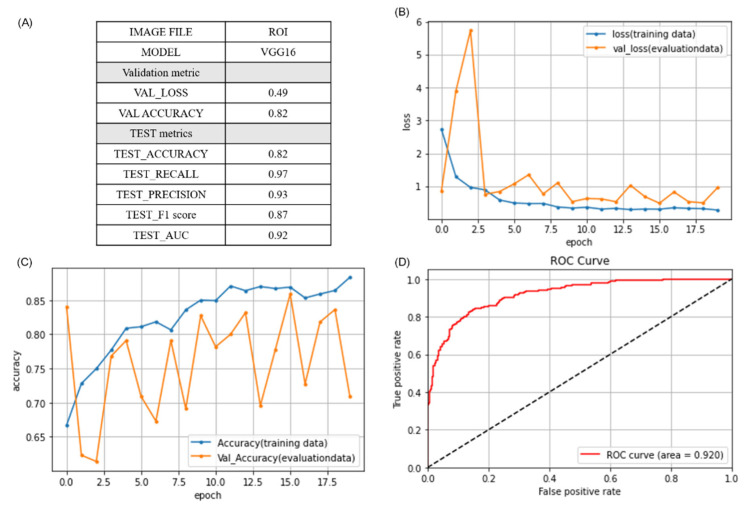
Performance details of VGG16 using ROI images: (**A**) performance metrics used to assess VGG16 using the VAL and TEST datasets; (**B**) loss learning curve of VGG16 indicating that the training data and validation data (VAL) both decreased to a point of stability with a small gap between the two, indicating good model fit; (**C**) accuracy learning curve of VGG16 revealing steady improvement in accuracy in each epoch using the training set during model fitting; (**D**) receiver operating characteristic (ROC) curve and the area under the curve (AUC) plotted using prediction data generated by the trained VGG16 model paired with the corresponding TEST set data, where an AUC of 0.92 indicates good model performance.

**Figure 5 diagnostics-11-01844-f005:**
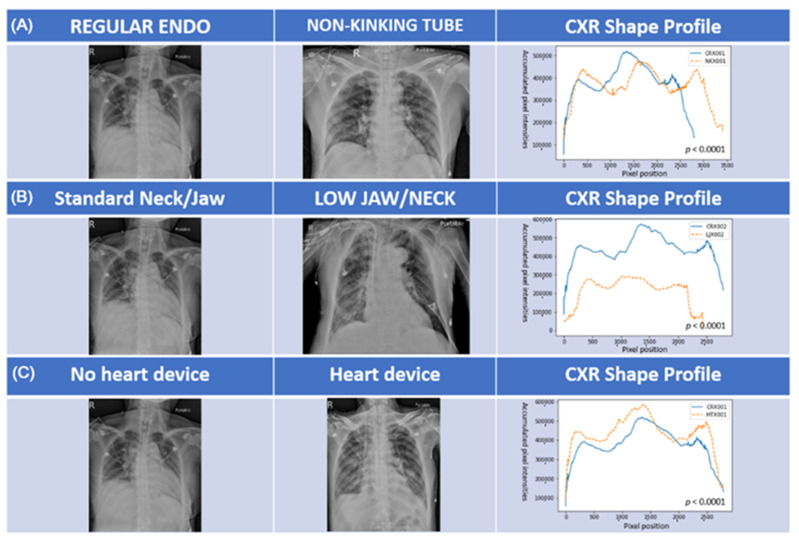
Analysis of the impact of the three major types of image noise using CXR shape profiles; (**A**) effect of tube material (tube factor); (**B**) effect of neck and jaw location (patient factor); (**C**) effect of heart device (environment factor).

**Table 1 diagnostics-11-01844-t001:** Total number of CXR images and distribution of data in this study.

CXR Type	FULL		ROI	
ETT position	CO	INCO	CO	INCO
Image NO.	2580 (60.1%)	1713 (39.9%)	2678 (61.7%)	1667 (38.3%)
Splitting				
TRAIN	1800 (60.04%)	1198 (39.96%)	2148 (61.72%)	1332 (38.28%)
VAL	500 (60.24%)	330 (39.76%)	405 (61.83%)	250 (38.17%)
TEST	280 (60.22%)	158 (39.78%)	134 (61.19%)	85 (38.81)

CO: correct position of endotracheal tube; INCO: incorrect position of endotracheal tube.

**Table 2 diagnostics-11-01844-t002:** Performance metrics used to assess the algorithms.

MODEL NAME	IMAGE FILE	ALGORITHM	VAL_LOSS	VAL_ACCURACY	TEST_ACCURACY	RECALL	PRECISION	F1 Score	AUC
FCNN_TPL20	FULL	CNN_TPL	1.05	0.57	0.53	0.63	0.61	0.62	0.49
RCNN_TPL20	ROI	CNN_TPL	1.04	0.75	0.55	0.66	0.63	0.64	0.54
FVG16TPL20	FULL	VGG16_TPL	0.60	0.59	0.61	0.77	0.65	0.70	0.55
RVG16TPL20	ROI	VGG16_TPL	0.62	0.60	0.72	0.96	0.70	0.81	0.82
FVG16_20	FULL	VGG16	0.81	0.63	0.61	1.00	0.86	0.76	0.56
RVG16_20	ROI	VGG16	0.49	0.82	0.82	0.97	0.93	0.87	0.92
FTLINCEPV3_20	FULL	Inception_V3	1.68	0.58	0.52	0.67	0.59	0.63	0.49
RTLINCEPV3_20	ROI	Inception_V3	0.70	0.61	0.61	1.00	0.61	0.76	0.50
FTLRESNT_20	FULL	ResNet50	2.56	0.60	0.56	0.79	0.60	0.68	0.47
RTLRESNT_20	ROI	ResNet50	2.24	0.68	0.48	0.55	0.58	0.56	0.46
FTLDENSENET_20	FULL	DenseNet169	2.77	0.48	0.48	0.57	0.59	0.49	0.50
RTLDENSENET_20	ROI	DenseNe169	1.44	0.78	0.58	0.82	0.62	0.70	0.57

Note: FULL: FULL group image; ROI: ROI group image; CNN: convolutional neural network; TPL: tensor projection layer; VGG16_TPL: VGG16 with tensor projection layer; VAL_LOSS: model loss using the validation (VAL) set; VAL_ACCURACY: model accuracy using the validation (VAL) set; TEST_ACCURACY: model accuracy using the TEST set; AUC: area under receiver operating characteristics curve.

## Data Availability

Not applicable.

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
