# Peer review of "Using Transfer Learning Method to Develop an Artificial Intelligence Assisted Triaging for Endotracheal Tube Position on Chest X-ray"

_diagnostics, 2021, doi:10.3390/diagnostics11101844_

Round 1

Reviewer 1 Report

The authors have examined 4,293 chest x-rays from 2,568 ICU patients to train and test an automated model for prediction of accurate endotracheal tube placement. They showed the benefit of transfer learning for this task. The methodology is well explained and soundly designed. The results are well presented. However, the manuscript will benefit from language editing and critical review.  

Author Response

Point 1: The authors have examined 4,293 chest x-rays from 2,568 ICU patients to train and test an automated model for prediction of accurate endotracheal tube placement. They showed the benefit of transfer learning for this task. The methodology is well explained and soundly designed. The results are well presented. However, the manuscript will benefit from language editing and critical review. 

Response 1:

We thank the reviewer for the valuable suggestion. We had English language editing and critical review by professional native English speakers for our manuscript, and we provide English editing certificate as follow.

Reviewer 2 Report

The paper was well-written. The idea of the paper is also interesting. However, the paper needs to be improved a little bit more prior.

1. In the Coding architecture Part 3 box of Figure 1, it is written Test accuracy. What does "Test accuracy" means? I could not find the description in the text.
2. Table 1 was separated into two pages.
3. Section 2.4. The python code structure was written in three modules: 1) algorithm-related, 2) image processing, and 3) model fitting and performance evaluation. Please provide a flowchart with a detailed explanation so that the reader understands the flow process of the algorithm.
4. Manual cropping presented in Figure 2 was operator-dependent. I believe the result was varied from one operator to another operators. How to avoid this issue? could the Authors provide an explanation?
5. Please add more content in the Conclusions section. The Conclusions must be linked to the Abstract.
6. The overall results of the study were outstanding. However, it would be better if there is a comparison with previous studies.
7. I noticed that the Authors used VGG16, could you please provide an architecture of the used method (VGG16) in the paper?

Author Response

Point 1:  In the Coding architecture Part 3 box of Figure 1, it is written Test accuracy. What does "Test accuracy" means? I could not find the description in the text.

Response 1:

The "Test accuracy" is the trained model's performance metric (accuracy) using the CXR image data in the Test set. The models in our study are established using the pre-trained models (VGG16, INCEPTION_V3, RESNET, and DENSENET169) and trained with the images in the "TRAINING SET and "VALIDATION SET." Then we use the "TEST SET" to evaluate the established models. The "TEST ACCURACY" results from established models using unseen data to provide an objective indicator of model performance.

Point 2:  Table 1 was separated into two pages.

Response 2:

We thank the reviewer for the valuable suggestions. However, Table 1 is not big, and it is not necessary to separate it into two pages. We can adjust the editing of the manuscript to present the whole Table 1 on one page.

Point 3:  Section 2.4. The python code structure was written in three modules: 1) algorithm-related, 2) image processing, and 3) model fitting and performance evaluation. Please provide a flowchart with a detailed explanation so that the reader understands the flow process of the algorithm.

Response 3:

We thank the reviewer for the valuable suggestions. We will provide a flowchart as a supplement file (Figures S1) with a detailed explanation of the Python code structure in our study to elucidate the flow process of the algorithm.

Point 4:  Manual cropping presented in Figure 2 was operator-dependent. I believe the result was varied from one operator to another operators. How to avoid this issue? Could the Authors provide an explanation?

Response 4:

We thank the reviewer for the valuable suggestions. There are indeed possible variations between operators when manual cropping, and we have had some designs to void this. First, we have the board-certificated intensivist who is very knowledgeable about CXR to do the cropping.  Besides, as mentioned in the method part of our study, we set a standardization about image cropping: (ROI) images including four landmarks: the right clavicle head, the left clavicle head, the carina, and the tip of the ETT (Figure 1). We use the two methods to reduce the variations between operators about image cropping.

Point 5:  Please add more content in the Conclusions section. The Conclusions must be linked to the Abstract.

Response 5:

We thank the reviewer for the valuable suggestions. We have revised the Conclusion section as below.

This study demonstrates the feasibility of using pre-trained models in developing a transfer learning-based AI system by which to automate the assessment of ETT placement in CXRs. Most of the pre-trained models performed better when using ROI images. The best performance was achieved using VGG16 in conjunction with ROI images (AUC= 92%, F1 score=0.87). Excessive image noise and inadequate hyperparameter tuning were the major causes of unsatisfactory performance. In such cases, TPL dimensionality reduction was shown to improve performance. Overall, these preliminary results demonstrate the feasibility of using transfer learning-based AI for the assessment of ETT placement in CXRs.

Point 6:  The overall results of the study were outstanding. However, it would be better if there is a comparison with previous studies.

Response 6:

We thank the reviewer for the valuable suggestions. In fact, we had compared several similar studies (using AI system for object detection and recognition in X-ray, not for disease diagnosis) in the Discussion Section (Page 8, line 267-278). And we noticed that some studies achieved better accuracy, which is up to 99.6%, therefore we also discussed some possible factors to improve our performance in the Discussion Section. 

Point 7:  I noticed that the Authors used VGG16, could you please provide an architecture of the used method (VGG16) in the paper?

Response 7:

We thank the reviewer for the valuable suggestions. We will provide a supplement file (Figures S2-S3) explaining the architecture of the used method (VGG16) in the revised manuscript.

Round 2

Reviewer 2 Report

Dear Authors,

Thank you for providing the revised version. I have read the revised documents. I have no further comments.

Kind regards,

- Reviewer -